# A Study of Performance Evaluation for Textile and Garment Enterprises

Chia-Nan Wang [1], Phuong-Thuy Thi Nguyen [1,*], Yen-Hui Wang [2,*] and Thanh-Tuan Dang [3]

1 Department of Industrial Engineering and Management, National Kaohsiung University of Science and Technology, Kaohsiung 807618, Taiwan
2 Department of Information Management, Chihlee University of Technology, New Taipei City 220305, Taiwan
3 Department of Logistics and Supply Chain Management, Hong Bang International University, Ho Chi Minh 72320, Vietnam
* Correspondence: phuongthuytn.iem@gmail.com (P.-T.T.N.); ttxyhw@mail.chihlee.edu.tw (Y.-H.W.)

**Abstract:** Vietnam's textile and garment enterprises make an important contribution to the country with the second largest export turnover. The existence and development of textile and garment enterprises have a significant influence on the socioeconomic development of Vietnam. Currently, Vietnam's textile and garment industry is facing difficulties caused by the COVID-19 pandemic, along with competition from foreign direct investment (FDI) enterprises. Therefore, it is imperative for managers to assess competitiveness by measuring their past and current performance indicators. This study assesses the performance of Vietnam's 10 textile and garment enterprises from 2017 to 2020 by combining the DEA–Malmquist productivity index (MPI) and epsilon-based measure (EBM) model. The proposed model considered three inputs (total assets, cost of goods sold, and liabilities) and two outputs (total revenue and gross profit). In addition to showing the best-performing companies from certain aspects during the period (2017–2020), the results show that the EBM method combined with the Malmquist model in the field can be successfully applied. This study is a reference for companies in the textile and garment industry to identify their position to improve their operational efficiency and overcome their weaknesses.

**Keywords:** efficiency; data envelopment analysis; Malmquist; EBM; textile and garment industry



## 1. Introduction

The textile industry involves the manufacturing of clothing and garments, embroidery, and the design, production, and distribution of yarns and fabrics. Vietnam's textile and garment enterprises make an important contribution to the country, with the second largest export turnover in 2018, as shown in Figure 1 [1,2].

According to the Vietnam Textile and Apparel Association (VITAS), the whole industry currently has more than 6000 enterprises with about three million employees, accounting for over 10% of the industry's workforce, growing at an annual rate of 18% since 2002 [3,4]. After China, Germany, and Bangladesh, Vietnam is one of the top four exporting countries of textiles and garments in the world [5]. In 2018, garment export turnover reached 28.78 billion USD [6]. According to VITAS data, the average annual growth rate of export value has been 14.74% over the last 5 years, with the export turnover in 2018 being 36.2 billion USD, accounting for 16% of the industrial production value of Vietnam. In the first 11 months of 2019, textile and garment exports reached 39.89 billion USD, up 7.8% over the same period last year. In 2020, the textile and garment industry was affected by the negative and prolonged effects of the COVID-19 epidemic. The industrial production index of the textile industry decreased by 0.5% [7].

Currently, Vietnam's textile and garment industry is facing difficulties and challenges. Firstly, Vietnamese textile and garment companies have to compete with foreign direct investment (FDI) companies. In 2007, Vietnam became a member of the World Trade

Organization (WTO). This attracted a number of foreign businesses to Vietnam to look for investment possibilities. FDI companies contributed up to 65% of export turnover, although they only accounted for 25% in volume [8]. Due to the growth potential of the market and the impact of a series of trade agreements, FDI inflows into the textile and garment sector have continuously increased. This not only brings many challenges to Vietnam's textile and garment industry, but also creates motivation for Vietnamese enterprises to increase quality, productivity, and technological innovation. In addition, foreign enterprises have modern machinery and technology, most of which are automated. Therefore, the cost is low, but the productivity is excellent. The export growth of Vietnamese textile and garment enterprises relies on outsourcing and low labor costs. However, these two items cannot be maintained for long. Outsourcing activities will shift to countries with lower labor costs, while labor costs in Vietnam are increasing. Furthermore, science and technology applications are still limited by the average level of technology, the low level of textile workers, etc. In short, the market requires businesses in the industry to continuously research and develop unique new products to attract customers.

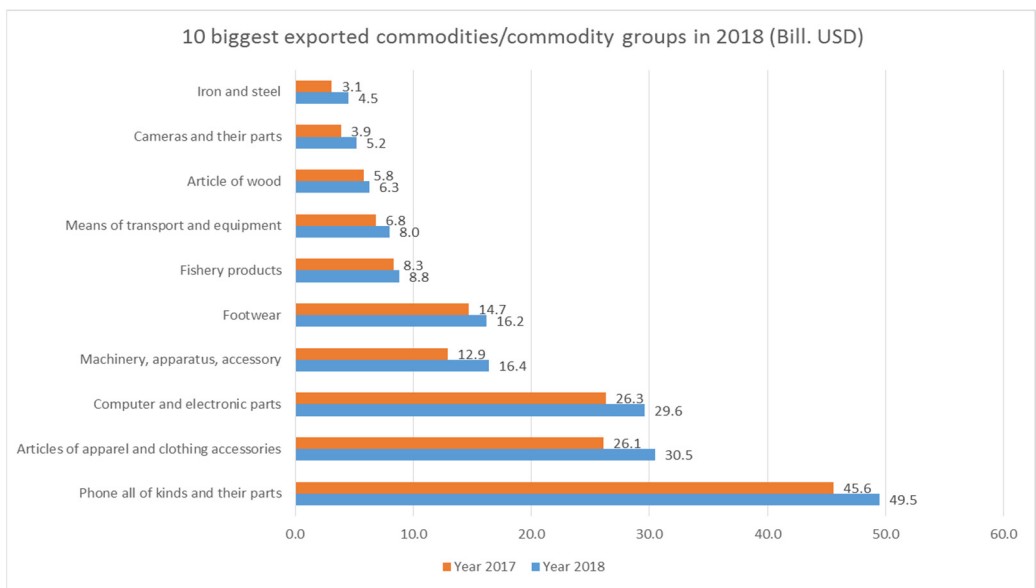

**Figure 1.** The ranking of exported commodities/commodity groups.

Secondly, because the US–China trade war affects the exchange rate between currencies, the price of processed goods in Vietnam is higher than that in some countries in the region such as Korea and China. This affects export orders. According to statistics from the Ministry of Industry and Trade, some businesses have only about 70% of new orders compared to the same period in 2018. If in 2018, by the middle of the year, many large enterprises had ordered until the end of the year, in 2019 they could only sign orders with small quantities on a monthly basis [9].

Thirdly, Vietnam's textile and garment industry has been heavily affected by the COVID-19 epidemic, due to the supply chain depending on a few key partners. The supply chain was interrupted when the COVID-19 epidemic broke out in early 2020 because Vietnam imported up to 89% of fabrics, with 55% from China, 16% from South Korea, 12% from Taiwan, and 6% from Japan, as shown in Figure 2 [10].

This makes businesses unable to meet the demand. The shutdown of the input manufacturing industry has led to a shortage of raw goods in Vietnam since January 2020. However, the drop in demand from the United States and Europe has led to cancellations of purchases, causing income and employment losses for domestic manufacturers. According to VITAS, textile and garment exports in the first 4 months of 2020 fell by 6.6% compared to previous years to 10.64 billion USD. Meanwhile, the total import value was 6.39 billion USD, down 8.76% compared to the same period last year. Since March 2020, 80% of textile

and garment factories have reduced shifts and rotated workers [7]. Furthermore, some businesses have had to implement social distancing and stop production, resulting in many workers losing their jobs or quitting.

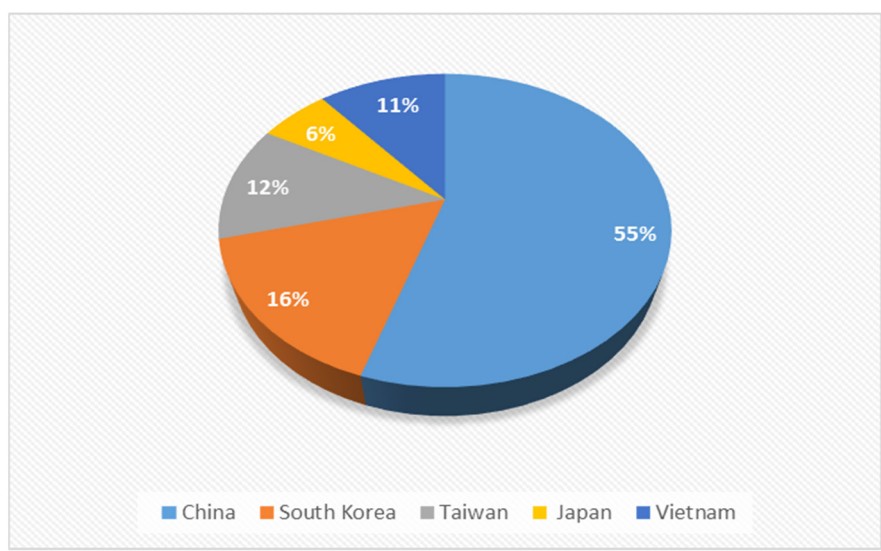

**Figure 2.** Imported fabric in Vietnam in 2019.

Vietnam's textile and garment is expanding and becoming increasingly essential to the country's economic development. The existence and development of textile and garment enterprises have a significant influence on the socioeconomic development of Vietnam. Therefore, managers should measure their historical and current performance efficiency indicators to assess their competitiveness and performance, especially at the present time when entrepreneurs have to face many difficulties and challenges as mentioned above.

The purpose of this paper is to evaluate the performance of textile companies in Vietnam. To help businesses have a more general view of the industry's activities, data envelopment analysis (DEA) seems to be a promising method. DEA has been used to assess the relative performance of organizational units known as decision-making units (DMUs) with input and output variables. The Malmquist model is used to evaluate the total productivity growth rate of DMUs, which can help businesses better understand the performance of Vietnam's textile and garment industry through changes in technical efficiency, technological progress, and total factor productivity. The epsilon-based measure (EBM) [11] is an epsilon-based DEA model that examines the efficiency and inefficiency of the DMU in a unified system that combines radial and non-radial models. This model is used to calculate the efficiency and inefficiency score, thereby ranking each company.

The literature on evaluating the performance of Vietnam's textile and garment industry is still limited. There are also no studies combining the DEA–Malmquist and EBM methods to evaluate the performance of Vietnam's textile and garment industry. This study assesses the performance of 10 Vietnamese textile and garment enterprises over the last four years, from 2017 to 2020 by combining the DEA–Malmquist productivity index (MPI) and EBM model. The main contributions of the paper are as follows: (1) the EBM method is proposed for the first time in combination with the DEA–Malmquist model to evaluate the performance of Vietnam's textile and garment industry; (2) this is a case study of 10 textile and garment enterprises in Vietnam used to illustrate the applicability of the proposed combined approach; (3) the results reflect that the EBM method combined with the Malmquist model in the field can be successfully applied.

This study is structured as follows: Section 1 is an introduction that summarizes the study, including the textile and garment industry's background, study motivation, and research process; Section 2 reviews the literature on the DEA method, specifically the MPI model, EBM model, and the textile and garment industry; Section 3 presents data sources

and determines the input and output for the DEA–Malmquist and EBM methods, along with results and analysis; Section 4 provides the discussion, conclusions, limitations, and future studies.

## 2. Literature Review

### 2.1. Data Envelopment Analysis (DEA) Model

Data envelopment analysis (DEA) is a mathematical programming-based approach to measuring relative efficiency across a large number of decision-making units (DMUs) and businesses with many inputs and outputs [12]. Charnes, Cooper, and Rhodes (CCR) introduced DEA in 1978 [12,13], which is based on Farrel's nonparametric method to evaluate technology efficiency [14,15]. Since then, many DEA models have been developed and used in different fields.

Many advancements have been made, with the goal of reducing the restrictions of earlier models, as well as the banker returns of variable rates, such as the Charnes and Cooper (BCC) model (1984) [13], which improves the constant returns to scale constraint of the CCR model. When dealing with excess input and output deficits, the slacks-based measure (SBM) model [16] evaluates the proportional change between inputs and outputs. In DEA, technical efficiency measurements are often classified into two types: radial measurements and non-radial measurements. Tone and Tsutsui [11] developed the EBM model in 2010 to address issues with radial and non-radial measurements of the ratio between input and output changes.

For example, the study by Drake and Hall [17] used DEA to examine the efficiency of scale and technique in Japanese banking; their finding showed that, in Japanese banking, accounting for the external impact of issue loans is crucial. Gómez and Cándido [18] used DEA to evaluate the efficiency of European health systems and found that people's health can be negatively impacted by high economic disparities in developed countries. Svetlana et al. [19] performed an efficiency evaluation of regional environmental management systems in Russia; the results showed that the efficiency of regional environmental management in many Russian regions has great potential for improvement. Wang et al. [20] evaluated the efficiency of LED energy enterprises in China using the DEA model. The results revealed that the total technical efficiency of China's LED energy firms has improved, owing to the LED energy enterprises' focus on technology advancement. Halkos and Nickolaos [21] utilized the DEA model to analyze the financial performance of enterprises participating in the Greek renewable energy sector; the results showed that enterprises producing wind energy outperform companies producing hydropower.

### 2.2. Malmquist Productivity Index (MPI)

According to Färe [22], the Malmquist method consists of two components: measurement of technical efficiency change and measurement of technological change. Many applications of DEA–Malmquist models can be found. For example, Xue et al. [23] used a three-stage DEA–Malmquist index approach to assess the static and dynamic efficiency of scientific research at HEIs in China; the results showed that the three-stage DEA model was more accurate than the classic DEA method for determining the efficiency of scientific research input and university output. The Malmquist meta-frontier analysis was used by Azad et al. [24] to assess the bank's efficacy in Malaysia. The results showed that Islamic banks performed best. Wang et al. [25] used the GM (1, 1) and DEA–Malmquist models to examine real estate company performance. The results demonstrated that, even though technology efficiency is steady throughout time, significant changes in efficiency in some organizations should be detected at the start of the process. Most businesses were able to maintain reasonably consistent productivity in the future. Wang and Li [26] also used the Malmquist model to analyze the carbon emissions performance of independent oil and natural gas producers in the United States. The results showed that independent oil and gas producers must invest more in emission-reduction techniques, such as energy

saving, leak detection and repair, outbreak reduction, and even renewable energy to ensure long-term success.

### 2.3. Epsilon-Based Measure (EBM)

Tone and Tsutsui [11] established the EBM model in 2010 to overcome problems with radial and non-radial measurements, as well as to examine the efficiency and inefficiency of DMUs in a single structure that incorporates radial measurement and non-radial measurements. In situations where the inputs or outputs might change proportionately, the radial measures of efficiency are evaluated; the non-radial measures, on the other hand, cause scaling and permit for independent changes of the respective slacks in the input or output. Mariano et al. [27] evaluated the performance of solar PV power plants in Taiwan using the EBM model. The top-performing DMU among nine selected solar PV plants was I (8200051) from Taiwan's southern region. The EBM model and gray forecasting were used by Wang et al. [28] to evaluate the efficiency of third-party logistics services. Wang et al. [29] used the EBM model and the LTS (A, A, A) model to assess the efficiency of the packaging industry's supply chain in Vietnam. The found that the packaging business as a whole has a high level of productivity. Wang et al. [30] used a combination of the DEA–Malmquist and EBM model to assess the performance of seaport operators. The results showed that the gap in the application of the EBM method in the marine industry was successfully bridged using the integrated framework.

### 2.4. Textile and Garment Industry Related Research

As mentioned above, the textile and garment industry plays a significant part in the daily life and economy of the country. The literature on the textile and garment industry's evaluation issues, particularly the efficiency assessment of textile and garment enterprises, is still scarce. There are also no studies applying EBM model to analyze the activities of textile and garment enterprises in Vietnam.

Some relevant research methods related to textile and garment are shown in Table 1. The DEA was used to assess performance, the CCR model was used to calculate the total efficiency score, the BBC model was used to assess the technical efficiency scale, and the SBM model was used to evaluate the efficiency ranking, as well as the delay problems of DMUs, in [8,31–35]. As mentioned in [36,37], the DEA window model was used by the original DEA model, which explored the trend of DMUs over time with multiple input and output variables; this is beneficial when dealing with small sample sizes. As can be shown in [35,36,38,39], to measure the change in total factor productivity, the Malmquist–DEA model was used, including changes in technical efficiency and change in technology change. Multicriteria decision models, such as the analytical hierarchy process (AHP) [40–42] and the technique for order of preference by similarity to ideal solution (TOPSIS) [43,44], are widely used to assess the performance of the textile and garment industry.

SBM, CCR, and BCC models are often utilized in relevant problems, as shown in the table below. On the other hand, the EBM model is not used in the decision making and evaluation process of the textile and garment industry. The EBM approach has the benefit of addressing weaknesses in the CCR, BCC, and SBM models. CCR and BCC models are radial techniques that focus on changes in input and output, respectively, while ignoring missing points. Meanwhile, CCR and BCC enhance SBM (non-radial), directly resolving errors but not those related to the corresponding change in input/output. As a result, the EBM model [11] was created as a solution to these flaws, including both radial and non-radial properties. The epsilon measure in an EBM model represents the diversity or dispersion of the observed dataset. In addition, slack represents the ability of less efficient units to improve their input and output variables relative to the standard target.

This omission caught the attention of the author. Therefore, the authors used the EBM model in combination with the DEA–Malmquist model to calculate the performance of 10 Vietnamese textile and garment companies.

**Table 1.** List of related methods of previous studies.

| No. | Authors | Year | DEA CCR | DEA BCC | DEA SBM | DEA Window | DEA Malmquist | (Fuzzy) TOPSIS | (Fuzzy) AHP |
|---|---|---|---|---|---|---|---|---|---|
| 1 | Chandra et al. [31] | 1998 | x | | | | | | |
| 2 | Jahanshahloo and Khodabakhshi [32] | 2004 | | x | | | | | |
| 3 | Joshi et al. [38] | 2010 | | | | | x | | |
| 4 | Zhao et al. [40] | 2011 | | | | | | | x |
| 5 | Zarbini et al. [40,43] | 2011 | | | | | | x | |
| 6 | Yayla et al. [44] | 2012 | | | | | | x | |
| 7 | Le et al. [33] | 2014 | | | x | | | | |
| 8 | Zhang et al. [39] | 2014 | | | | | x | | |
| 9 | Jakhar [41] | 2015 | | | | | | | x |
| 10 | Wang et al. [8] | 2017 | | | x | | | | |
| 11 | Le and Wang [36] | 2017 | | | | x | x | | |
| 12 | Guarnieria and Trojan [42] | 2018 | | | | | | | x |
| 13 | Mehmet et al. [37] | 2019 | | | | x | | | |
| 14 | Tran et al. [34] | 2021 | x | x | | | | | |
| 15 | Nguyen and Vu [35] | 2021 | | | x | | x | | |

## 3. Materials and Methods

### 3.1. Research Framework

In this study, the DEA–Malmquist and EBM models were used to evaluate the efficiency of 10 Vietnamese textile and garment enterprises from 2017 to 2020. As shown in Figure 3, the research process could be divided into two phases. In the first phase, the DEA–Malmquist model was used to productivity index analysis to evaluate total productivity changes resulting from technical efficiency change (catch-up) and technological efficiency change (frontier-shift) with inputs (total assets, cost of goods sold, and liabilities) and outputs (total revenue and gross profit). Before applying the DEA–Malmquist model, Pearson correlation was used to evaluate the correlation between input and output variables. In the second phase, the EBM model was used to calculate the efficiency and inefficiency score of DMUs. Before using the EBM model, the diversity and affinity coefficient indices were verified.

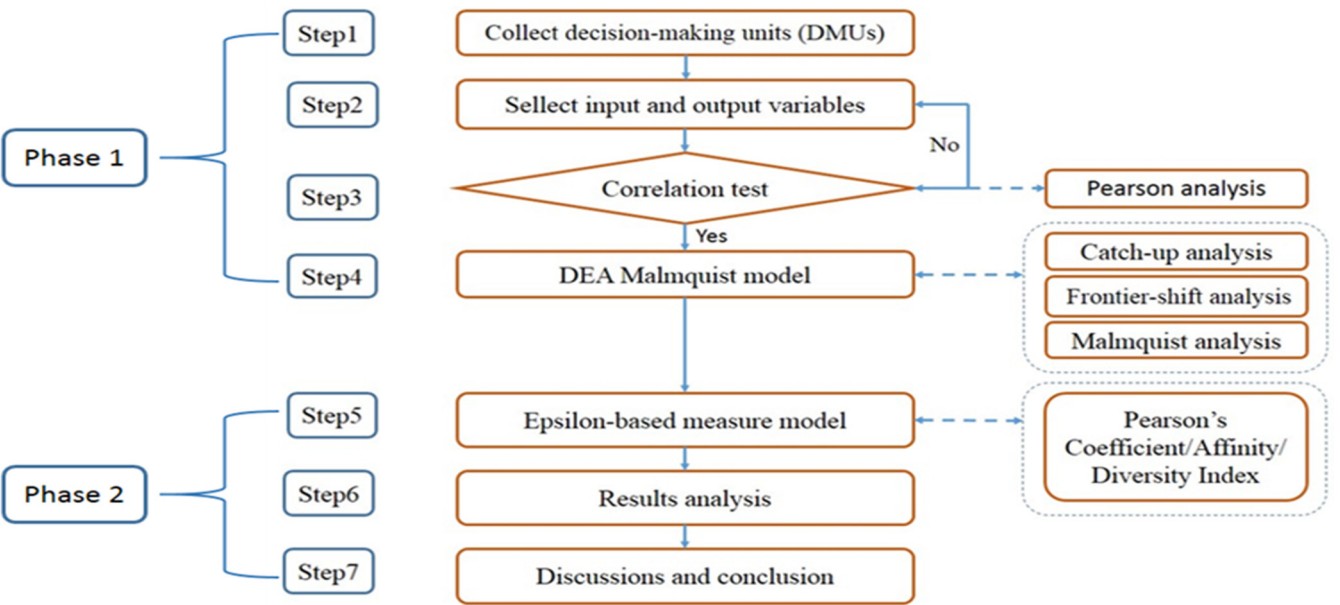

**Figure 3.** Research framework.

### 3.2. DMU Selection

There are many textile and garment enterprises in Vietnam with different production scales, products, and techniques. Therefore, the authors researched and selected the leading enterprises in the textile and garment industry based on the reports of financial experts, i.e., Vietnam Credit [45], and business reports from reputable websites [46,47]. However, accessing all of a company's data is difficult because not all of them publish financial statements publicly. Furthermore, enterprises with negative values in their financial accounts are excluded from this study. The authors selected the top 10 textile and garment enterprises in Vietnam that were listed in Viet stock [48] from 2017 to 2020. Table 2 shows a list of all companies.

**Table 2.** DMUs list.

| DMU | Symbol | Companies Name | Stock Code |
|---|---|---|---|
| DMU1 | Viet Tien | Viet Tien Garment JSC | VGG |
| DMU2 | Phong Phu | Phong Phu JSC | PHONG PHU CORP |
| DMU3 | Ha Noi | Hanoi Textile and Garment JSC | HANOSIMEX |
| DMU4 | Song Hong | Song Hong Garment JSC | MSH |
| DMU5 | Garment 10 | Garment Corporation 10 JSC | M10 |
| DMU6 | Binh Thanh | Binh Thanh Manufacturing, Trading and Import | GIL |
| DMU7 | Thanh Cong | Thanh Cong Textile—Investment—Trading | TCM |
| DMU8 | TNG | TNG Investment and Trading JSC | TNG |
| DMU9 | Sai Gon | Garmex Saigon JSC | GMC |
| DMU10 | TDT | TDT Investment and Development JSC | TDT |

### 3.3. Input and Output Selection

As shown in Figure 4, this study analyzes three input variables (total assets, cost of goods sold, and liabilities) and two output factors (total revenue and gross profit) on the basis of the inputs and outputs utilized in previous relevant research (as shown in Table 3 below).

(I1) Total assets are all of the resources that a company owns and manages, including both current and long-term assets.

(I2) Cost of goods sold is the inventory value of goods sold for a specific period of time.

(I3) Liabilities are the enterprise's liabilities stemming from previous events and transactions, which the company has to pay with its resources.

(O1) Total revenue is the total amount money earned by a corporation through the sale of its goods or services over time (a day, a week, a month, or a year).

(O2) Gross profit is the portion of profit a company earns after deducting the costs involved in making and selling the product or the costs involved in providing the company's services.

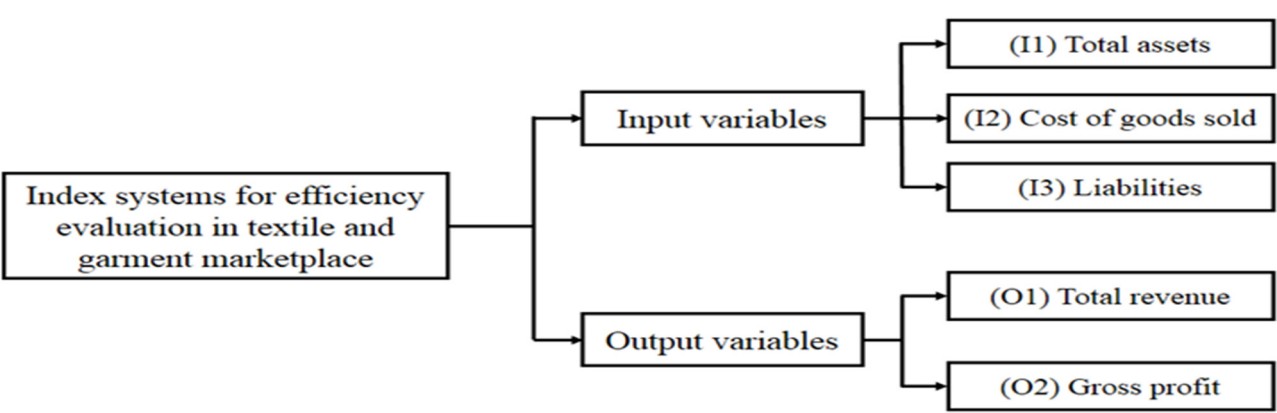

**Figure 4.** Selection of inputs and outputs.

**Table 3.** Input and output variables used in related research.

| Author | Input Factors | Output Factors |
|---|---|---|
| Wang et al. [8] | Total assets<br>Cost of sold capital<br>Selling expenses<br>General and administration expenses | Revenue of sales<br>Profit after tax |
| Nham and Wang [33] | Fixed assets<br>Capital<br>Operating expenses | Net sales<br>Earnings per share |
| Tran et al. [34] | The average number of employees per month<br>The wage fund<br>Total capital<br>Total cost | Total revenue<br>Gross profit |
| Wang et al. [30] | Total assets<br>Owner's equity<br>Liabilities<br>Operating expense | Total revenue<br>Gross profit |
| Wang et al. [49] | Total assets<br>Equity<br>Total liabilities<br>Cost of sales | Total revenue<br>Gross profit |
| Wang et al. [29] | Total assets<br>Cost of goods sold<br>Operating expenses | Total revenue<br>Gross profit |

*3.4. Data Envelopment Analysis (DEA)*

3.4.1. Pearson Correlation

Before using the DEA model, we must test the Pearson correlation to make sure that the input and output variables have an isotonic relationship. The strength of a relationship between two variables is measured using Pearson's correlation coefficient; correlation coefficients are always in the range of −1 to +1. When the correlation coefficient is close to ±1, the two groups are getting closer to having a perfect linear relationship [30].

Pearson's correlation coefficient is expressed as follows [25]:

$$r_{xy} = \frac{\sum_{i=1}^{n}(x_i - \bar{x})(y_i - \bar{y})}{\sqrt{\sum_{i=1}^{n}(x_i - \bar{x})^2 \sum_{i=1}^{n}(y_i - \bar{y})^2}}, \tag{1}$$

where *n* is the size of the sample, and $x_i$, $y_i$ denote the individual sample points related to *i* [49].

The correlation coefficients are described in detail in Table 4 [25].

**Table 4.** Pearson correlation.

| Correlation | Degree of Correlation |
|---|---|
| >0.8 | Very high |
| 0.6–0.8 | High |
| 0.4–0.6 | Medium |
| 0.2–0.4 | Low |
| <0.2 | Very low |

3.4.2. DEA–Malmquist Model

The Malmquist productivity index (MPI) [22] is the product of the change in technological progress and the change in technical efficiency. MPI can be calculated using the multiplying technical efficiency change (catch-up index) and technological change (frontier-shift index) [25].

The $DMU_i$ determined at period 1 is $(x_i^1, y_i^1)$ and at period 2 is $(x_i^2, y_i^2)$. The efficiency score of the DMU$_i$ $(x_i^1, y_i^1)^{t_1}$ was measured using the technological frontier $t_2$: $d^{t_2}\left((x_{i,}y_i)^{t_1}\right)(t_1 = 1, \, 2 \text{ and } t_2 = 1, \, 2)$ [49].

The following formulas can be used to calculate the catch-up index (CA), frontier-shift index (FR), and MPI [49]:

$$CA = \frac{d^2((x_i, y_i)^2)}{d^1((x_i, y_i)^1)}. \tag{2}$$

$$FR = \left[\frac{d^1((x_i, y_i)^1)}{d^2((x_i, y_i)^1)} \times \frac{d^1((x_i, y_i)^2)}{d^2((x_i, y_i)^2)}\right]^{\frac{1}{2}}. \tag{3}$$

$$MI = \left[\frac{d^1((x_i, y_i)^2)}{d^1((x_i, y_i)^1)} \times \frac{d^2((x_i, y_i)^2)}{d^2((x_i, y_i)^1)}\right]^{\frac{1}{2}}. \tag{4}$$

The results of the MPI are divided into three cases [25]:

(1)　MPI > 1: productivity improvement.
(2)　MPI = 1: constant productivity.
(3)　MPI < 1: decrease in productivity.

### 3.4.3. Epsilon-Based Measure Efficiency

In DEA, there are two types of technical efficiency measures: radial measures and non-radial measures. The radial measurement takes only the corresponding change in input or output, ignoring any slack. The non-radial measurement, on the other hand, deals directly with slacks and is unconcerned about the proportion of inputs and outputs changing. As a result, in some situations, both can lead to incorrect evaluation. To address this problem, the EBM model was created. Both radial and non-radial features are combined in the model. This framework contains two parameters, one scalar and one vector, which are defined by affinity index in relation to the inputs and outputs. These two factors are used to combine the radial and non-radial models into a single model for evaluating DMU efficiency [29].

By showing that the EBM is input-oriented (EBM I-C) for $DMU_0 = (x_0, \, y_0)$, we then calculate it as follows [29]:

$$\gamma^* = \min_{\theta, \lambda, s^-} \theta - \varepsilon_x \sum_{i=1}^{s} \frac{w_i^- s_i^-}{x_{i0}}. \tag{5}$$

This is subject to

$$\theta_{x0} - X\lambda - s^- = 0,$$

$$Y\lambda \geq y_0, \, \lambda \geq 0, \, s^- \geq 0,$$

where the weight (relative importance) of input (*i*) is $w_i^-$ and $\sum_{i=1}^{s} w_i^- = 1 \left(w_i^- \geq 0 \forall i\right)$, and $\varepsilon_x$ is the parameter that integrates the radial $\theta$ and non-radial slacks terms.

### Diversity Index and Affinity Index

Pearson's correlation is important in the DEA because it clarifies the association between two variables. It converts the raw data into a correlation estimate. If the Pearson index is high, it indicates that the two variables are related. On the other hand, a low correlation coefficient indicates a skewed input–output relationship. Pearson's correlation coefficient is a number that varies from $-1$ to $+1$ [29].

Furthermore, one of the most essential aspects in a DEA is weight. The weight determines the effect of the input on the output [50]. If the weight is close to 0, it means that the output is unaffected by changes in the input. Positive weights suggest an inverse relationship between input and output, i.e., if the input increases, the output decreases.

The values of $\varepsilon_x$ and $w_i$ have a significant impact on determining the efficiency of DMUs in the EBM model. The EBM model, on the other hand, employs the affinity index between two vectors rather than the Pearson's correlation coefficient as a model.

Let $a \in R_+^n$ and $b \in R_+^n$ be two non-negative vectors with $a$ dimension $n$. They show the values that have been checked for a certain input component in $n$ DMUs. S $(a, b)$ is the affinity index between vectors $a$ and $b$ that has the following characteristics [29]:

$$S\ (a,a) = 1(\forall_a)\ \text{Identical.}$$
$$S(a,b) = S(a,b)\ \text{Symmetric.}$$
$$S(ta,b) = S(a,b)/(\forall t > 0)\ \text{Unit} - \text{invariant.}$$
$$1 \geq S(a,b) \geq 0/(\forall a,b).$$

Let us define

$$c_j = ln\frac{b_j}{a_j} = (j = 1, \ldots, n),$$

$$\bar{c} = \frac{1}{n}\sum_{j=1}^{n} c_j, \tag{6}$$

$$c_{\max} = \underset{j}{\max}\{c_j\}, c_{\min} = \underset{j}{\min}\{c_j\}.$$

The diversity index of vectors $(a, b)$ is calculated as the divergence of $\{c_j\}$ from the average $\bar{c}$:

$$D(a,b) = \frac{\sum_{j=1}^{n}\left| c_j - \bar{c} \right|}{n(c_{max} - c_{min})} = 0\ if\ c_{max} = c_{min}, \tag{7}$$

and $0 \leq D(a,b) = D(b,a) \leq \frac{1}{2}$.

$D(a,b) = 0$ only if vector $a$ and vector $b$ are proportional.

If we denote the affinity index between vector $a$ and vector $b$ as $S(a,b)$, then

$$S(a,b) = 1 - 2D(a,b). \tag{8}$$

If $1 \geq S(a,b) \geq 0$, $S(a,b)$ is accomplished with Properties (5) and (6).

Pearson's correlation coefficient $(P(a,b))$ in DEA is determined using the following equation:

$$P(a,b) = \frac{\sum_{j=1}^{n}\left(a_j - \bar{a}\right)\left(b_j - \bar{b}\right)}{\sum_{j=1}^{n}\left(a_j - \bar{a}\right)^2\left(b_j - \bar{b}\right)^2}, \tag{9}$$

where $\bar{a}$ and $\bar{b}$ are the average of $a_j$ and $b_j$, respectively.

The affinity index, on the other hand, replaces Pearson's correlation coefficient $(P(a, b))$ in the EBM model. Pearson's index range is $-1 \leq P(a,b) \leq 1$. As a result, there is no guarantee that the major vector only contains positive components when assessing the fundamental factor. As a result, it is set to $0 \leq P(a,b) \leq 1$ [29].

## 4. Results Analysis

### 4.1. Data Analysis

The following tables present the results of textile and garment enterprises from 2017 to 2020. Total assets, cost of goods sold, and liabilities were the input variables, while total revenue and gross profit were the output variables. The practical data from 2017 to 2020 collected from Viet Stock.Vn Website [48] are shown in Table 5.

**Table 5.** The financial results of textile and garment companies in 2017–2020.

| | | Data in 2017 (Currency Unit: Million USD) | | | |
|---|---|---|---|---|---|
| **DMUs** | **(I) Total Assets** | **(I) Cost of Goods Sold** | **(I) Liabilities** | **(O) Total Revenue** | **(O) Gross Profit** |
| DMU1 | 4,249,750 | 7,464,275 | 2,798,007 | 8,458,166 | 987,616 |
| DMU2 | 5,311,729 | 2,734,374 | 3,661,196 | 3,024,185 | 286,249 |
| DMU3 | 2,304,447 | 2,127,647 | 1,892,494 | 2,360,751 | 220,559 |
| DMU4 | 2,380,600 | 2,717,910 | 1,625,380 | 3,282,451 | 563,976 |
| DMU5 | 1,364,529 | 2,584,207 | 995,396 | 3,028,555 | 443,800 |
| DMU6 | 1,487,143 | 1,816,545 | 927,325 | 2,169,958 | 353,414 |
| DMU7 | 3,035,382 | 2,706,189 | 1,963,763 | 3,209,692 | 502,881 |
| DMU8 | 2,225,690 | 2,051,588 | 1,596,422 | 2,491,019 | 437,019 |
| DMU9 | 908,284 | 1,344,066 | 613,554 | 1,610,475 | 260,982 |
| DMU10 | 209,183 | 170,869 | 114,868 | 217,062 | 45,713 |
| | | Data in 2018 (Currency Unit: Million USD) | | | |
| **DMUs** | **(I) Total Assets** | **(I) Cost of Goods Sold** | **(I) Liabilities** | **(O) Total Revenue** | **(O) Gross Profit** |
| DMU1 | 4,701,038 | 8,546,828 | 3,031,269 | 9,719,646 | 1,170,171 |
| DMU2 | 5,427,848 | 3,204,732 | 3,746,469 | 3,509,968 | 294,578 |
| DMU3 | 2,510,675 | 2,287,968 | 1,943,307 | 2,558,537 | 257,531 |
| DMU4 | 2,520,977 | 3,157,345 | 1,587,254 | 3,950,894 | 793,482 |
| DMU5 | 1,569,492 | 2,513,677 | 1,194,869 | 2,980,318 | 466,347 |
| DMU6 | 1,842,965 | 1,877,858 | 1,134,056 | 2,253,631 | 375,773 |
| DMU7 | 3,247,326 | 2,983,240 | 1,970,928 | 3,664,445 | 678,771 |
| DMU8 | 2,595,435 | 2,971,920 | 1,801,371 | 3,612,897 | 640,977 |
| DMU9 | 1,010,674 | 1,675,340 | 630,076 | 2,045,323 | 363,560 |
| DMU10 | 250,179 | 224,812 | 144,850 | 286,193 | 60,726 |
| | | Data in 2019 (Currency Unit: Million USD) | | | |
| **DMUs** | **(I) Total Assets** | **(I) Cost of Goods Sold** | **(I) Liabilities** | **(O) Total Revenue** | **(O) Gross Profit** |
| DMU1 | 4,982,865 | 7,906,892 | 2,986,637 | 9,037,020 | 1,128,667 |
| DMU2 | 4,535,136 | 3,045,489 | 2,994,898 | 3,350,394 | 290,202 |
| DMU3 | 2,144,743 | 2,256,100 | 1,603,087 | 2,420,818 | 147,829 |
| DMU4 | 2,566,212 | 3,482,815 | 1,330,468 | 4,411,872 | 928,438 |
| DMU5 | 1,588,021 | 2,838,517 | 1,196,952 | 3,351,258 | 512,319 |
| DMU6 | 1,898,449 | 2,158,896 | 1,061,974 | 2,538,355 | 379,459 |
| DMU7 | 2,922,805 | 3,065,482 | 1,497,538 | 3,645,053 | 578,718 |
| DMU8 | 3,027,410 | 3,825,318 | 1,960,689 | 4,617,542 | 786,906 |
| DMU9 | 1,028,988 | 1,454,755 | 545,563 | 1,749,298 | 293,016 |
| DMU10 | 340,830 | 284,522 | 185,807 | 366,130 | 80,481 |

| | | Data in 2020 (Currency Unit: Million USD) | | | |
|---|---|---|---|---|---|
| **DMUs** | **(I) Total Assets** | **(I) Cost of Goods Sold** | **(I) Liabilities** | **(O) Total Revenue** | **(O) Gross Profit** |
| DMU1 | 4,736,189 | 6,450,347 | 2,823,291 | 7,123,237 | 670,612 |
| DMU2 | 3,780,226 | 1,859,226 | 2,149,688 | 2,106,567 | 239,908 |
| DMU3 | 1,806,969 | 1,209,500 | 1,271,631 | 1,344,824 | 115,786 |
| DMU4 | 2,627,755 | 3,062,365 | 1,185,555 | 3,817,925 | 751,044 |
| DMU5 | 1,588,766 | 2,978,495 | 1,193,577 | 3,453,925 | 468,808 |
| DMU6 | 2,708,562 | 2,820,903 | 1,418,574 | 3,456,745 | 635,842 |
| DMU7 | 2,976,423 | 2,849,534 | 1,337,688 | 3,470,466 | 620,183 |
| DMU8 | 3,554,955 | 3,804,243 | 2,406,975 | 4,480,200 | 675,957 |
| DMU9 | 1,222,790 | 1,272,030 | 564,362 | 1,474,983 | 202,537 |
| DMU10 | 394,735 | 195,021 | 224,775 | 272,099 | 75,808 |

*4.2. Pearson Correlation Check*

According to the Pearson coefficient correlation values in the table below, the correlation between input and output variables in this study was positive and significant at the 0.01 level (Table 6). From the results, all coefficients were statistically significant, and the range of Pearson values was from 0.44 to 1. This means that the data utilized meet isotropic requirements and can be used in DEA calculations. As a result, the input and output variables chosen were appropriate for analyzing and evaluating the performance of Vietnamese textile and garment enterprises.

**Table 6.** Pearson correlation coefficients from 2017–2020.

| | | Total Assets (TA) | Cost of Goods Sold (CGS) | Liabilities (L) | Total Revenue (TR) | Gross Profit (GP) |
|---|---|---|---|---|---|---|
| 2017 | TA | 1 | 0.6682 | 0.9929 | 0.6555 | 0.5182 |
| | CGS | 0.6682 | 1 | 0.6374 | 0.9991 | 0.9289 |
| | L | 0.9929 | 0.6374 | 1 | 0.6227 | 0.4706 |
| | TR | 0.6555 | 0.9991 | 0.6227 | 1 | 0.9439 |
| | GP | 0.5182 | 0.9289 | 0.4706 | 0.9439 | 1 |
| 2018 | TA | 1 | 0.7207 | 0.9910 | 0.7045 | 0.5115 |
| | CGS | 0.7207 | 1 | 0.6836 | 0.9981 | 0.8761 |
| | L | 0.9910 | 0.6836 | 1 | 0.6629 | 0.4430 |
| | TR | 0.7045 | 0.9981 | 0.6629 | 1 | 0.9040 |
| | GP | 0.5115 | 0.8761 | 0.4430 | 0.9040 | 1 |
| 2019 | TA | 1 | 0.8359 | 0.9762 | 0.8183 | 0.6106 |
| | CGS | 0.8359 | 1 | 0.7873 | 0.9972 | 0.8611 |
| | L | 0.9762 | 0.7873 | 1 | 0.7606 | 0.5045 |
| | TR | 0.8183 | 0.9972 | 0.7606 | 1 | 0.8965 |
| | GP | 0.6106 | 0.8611 | 0.5045 | 0.8965 | 1 |
| 2020 | TA | 1 | 0.8137 | 0.9503 | 0.8098 | 0.6328 |
| | CGS | 0.8137 | 1 | 0.8086 | 0.9967 | 0.7921 |
| | L | 0.9503 | 0.8086 | 1 | 0.7922 | 0.5333 |
| | TR | 0.8098 | 0.9967 | 0.7922 | 1 | 0.8389 |
| | GP | 0.6328 | 0.7921 | 0.5333 | 0.8389 | 1 |

*4.3. Results of Malmquist Model*

The Malmquist productivity index (MPI) was utilized to assess the performance of 10 DMUs as a function of technical efficiency change (catch-up index) and technological

change (frontier-shift index). The results are divided into three sections: catch-up, frontier-shift, and Malmquist.

### 4.3.1. Technical Efficiency Change (Catch-Up Index—CA)

In the period 2017–2020, the catch-up index (CA) was used to assess the technological efficiency of Vietnamese textile and garment firms. A CA less than 1 indicates that the index has deteriorated, while a CA greater than 1 indicates that the index has improved.

The CA of the 10 DMUs from 2017 to 2020 is shown in Table 7 and Figure 5.

**Table 7.** Catch-up index of DMUs (2017–2020).

| Catch-Up | 2017 ≥ 2018 | 2018 ≥ 2019 | 2019 ≥ 2020 | Average |
|---|---|---|---|---|
| DMU1 | 1.0576 | 0.9533 | 0.8735 | 0.9614 |
| DMU2 | 1.0228 | 1.0865 | 0.9612 | 1.0235 |
| DMU3 | 1.0037 | 1.0733 | 0.8508 | 0.9760 |
| DMU4 | 1.1387 | 1.2597 | 0.9552 | 1.1179 |
| DMU5 | 0.8086 | 1.1892 | 1.0914 | 1.0297 |
| DMU6 | 0.8070 | 1.1033 | 1.0674 | 0.9926 |
| DMU7 | 1.0428 | 1.0719 | 1.0654 | 1.0600 |
| DMU8 | 0.9549 | 1.1081 | 0.8925 | 0.9852 |
| DMU9 | 1.1339 | 0.8480 | 0.8471 | 0.9430 |
| DMU10 | 0.9176 | 0.9955 | 1.1711 | 1.0281 |
| Average | 0.9888 | 1.0689 | 0.9776 | 1.0117 |
| Max | 1.1387 | 1.2597 | 1.1711 | 1.1179 |
| Min | 0.8070 | 0.8480 | 0.8471 | 0.9430 |
| SD | 0.1175 | 0.1162 | 0.1146 | 0.0516 |

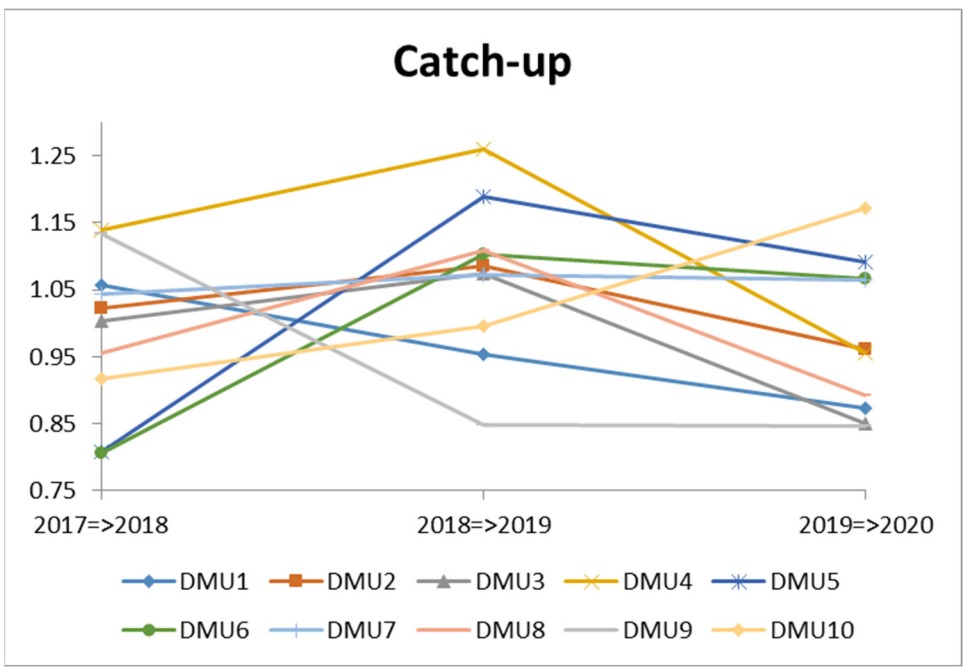

**Figure 5.** The technical efficiency change (catch-up) of DMUs (2017–2020).

In overall, the technological efficiency of all DMUs fluctuated during the 2017–2020 timeframe. During this time period, the average CA of all DMUs was 1.0117. With

CA = 1.1179, DMU4 (Song Hong) had the best efficiency performance, while DMU9 (Sai Gon) had the worst performance with CA = 0.9430.

Specifically, six of the 10 DMUs attained technical efficiency (average CA > 1) in the period 2017–2018. DMU4 (Song Hong) had the best results, with a CA of 1.1387. DMU6 (Binh Thach), on the other hand, had the lowest performance, with CA = 0.8070.

During the years 2018–2019, there was an increase in the technological efficiency improvement of enterprises compared to the years 2017–2018. Seven out of 10 DMUs were technically efficient (average CA > 1). DMU4 (Red River), the most popular DMU, had the best performance, with CA = 1.2597. Meanwhile, with CA = 0.8480, DMU9 (Sai Gon) had the lowest technical efficiency. Surprisingly, the DMUs that had low performance in 2017–2018, with scores below 1 (such as DMU6—Binh Thach, DMU8—TNG, and DMU5—Garment 10), improved dramatically in 2018–2019, with scores better than 1.

In the 2019–2020 period, CA started to drop with a regressive score of 0.9776. Only four out of 10 DMUs achieved the best performance (average CA > 1). DMU10 (TDT) achieved the highest CA (1.1711), while DMU9 (Sai Gon) had the lowest performance with CA = 0.8471. In particular, DMU10 was always in the group with a low CA index in the previous stages, instead showing a great improvement when reaching the highest efficiency in this period. In contrast, some companies showed signs of a decline in technical efficiency, such as DMU4 (Song Hong), DMU2 (Phong Phu), and DMU3 (Ha Noi). As a result, enterprises with weak results should focus on their technical aspects in order to improve their market competitiveness.

### 4.3.2. Technological Change (Frontier-Shift Index—FR)

To evaluate the technological efficiency or the effective frontier of 10 DMUs, the frontier-shift index (FR) was applied. Investment in production technology will improve labor productivity and promote the competitiveness of businesses in the same industry. Many of Vietnam's textile and garment enterprises have produced high-quality items as a result of innovative research and technology.

Table 8 and Figure 6 show the technological efficiency (frontier-shift) of the 10 DMUs from 2017 to 2020.

**Table 8.** Technological change of DMUs (2017–2020).

| Frontier | 2017 $\geq$ 2018 | 2018 $\geq$ 2019 | 2019 $\geq$ 2020 | Average |
|:---:|:---:|:---:|:---:|:---:|
| DMU1 | 1.0078 | 0.9627 | 0.9777 | 0.9827 |
| DMU2 | 1.0199 | 0.9865 | 0.9697 | 0.9921 |
| DMU3 | 1.0110 | 0.9727 | 0.9870 | 0.9902 |
| DMU4 | 1.1670 | 1.0156 | 0.9213 | 1.0346 |
| DMU5 | 1.0772 | 0.9555 | 0.9507 | 0.9945 |
| DMU6 | 1.1355 | 0.9733 | 0.9592 | 1.0227 |
| DMU7 | 1.1362 | 0.9992 | 0.9558 | 1.0304 |
| DMU8 | 1.2057 | 0.9685 | 0.9668 | 1.0470 |
| DMU9 | 1.1009 | 1.0283 | 0.9656 | 1.0316 |
| DMU10 | 1.1095 | 1.1058 | 0.9987 | 1.0713 |
| Average | 1.0971 | 0.9968 | 0.9652 | 1.0197 |
| Max | 1.2057 | 1.1058 | 0.9987 | 1.0713 |
| Min | 1.0078 | 0.9555 | 0.9213 | 0.9827 |
| SD | 0.0680 | 0.0449 | 0.0212 | 0.0289 |

In general, the average FR score for all DMUs was higher than 1 (FR = 1.0197), indicating that they have progressed technologically. During 2017–2020, most DMUs showed

common trends, increasing from 2017 to 2018, but severely declining from 2018–2019 to 2019–2020.

During the 2017–2018 period, all DMUs had an FR of more than one (average FR > 1), showing that textile and garment industry worked hard to develop and innovate technology and had positive outcomes. The highest score was 1.2057 for DMU8 (TNG), while the lowest score was 1.0078 for DMU1 (Viet Tien). This period's average technological efficiency yielded a progressive score of 1.0971. However, they were unable to maintain their position in the next period (2018–2019), as most FR values dropped under 1, with the exception of DMU4 (Song Hong), DMU9 (Sai Gon), and DMU10 (TDT).

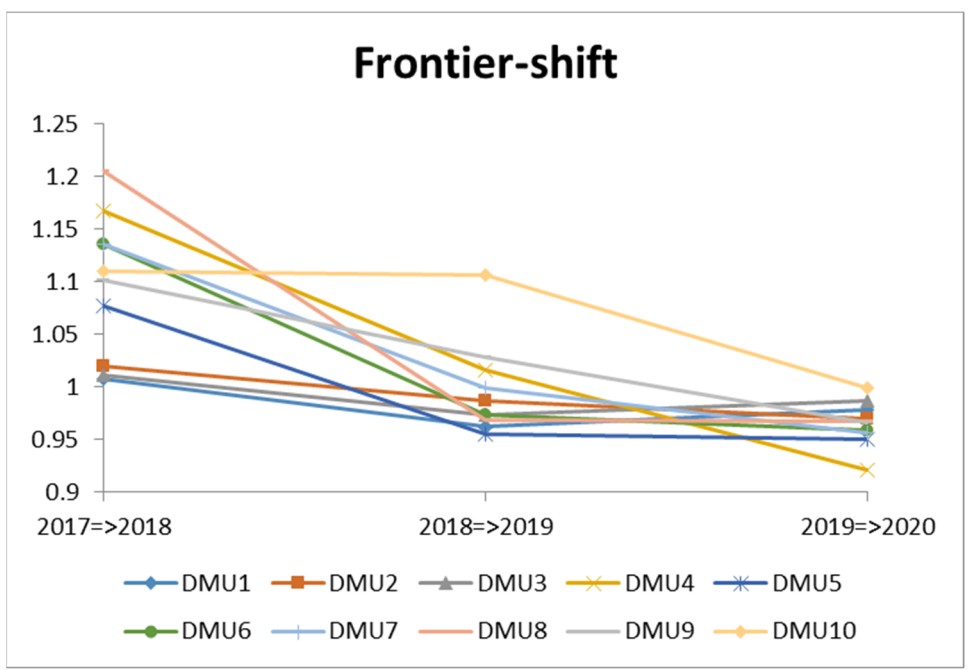

**Figure 6.** Technological change (frontier-shift) of DMUs (2017–2020).

FR began to decline in the 2018–2019 period, with a regressive score of 0.9968. Only three out of 10 DMUs achieved technological efficiency. DMU10 (TDT) achieved the highest FR (1.1058), while DMU5 (Garment 10) had the lowest technological efficiency with FR = 0.9555. This suggests that, during this time period, all DMUs were not highly technologically efficient.

Figure 6 shows that the majority of textile and garment enterprises were not able to retain their efficiency in the next period (2019–2020). All of the DMUs exhibited poor technological performance (FR < 1), with DMU4 (Song Hong) having the worst performance (FR = 0.9213). Manufacturers' technological efficiency decreased during this time period.

Therefore, enterprises must focus on complete investment, particularly in terms of technology, in order to improve operational efficiency and keep up with the present textile and garment industry growth.

### 4.3.3. Malmquist Productivity Index (MPI)—Total Productivity Change

The MPI was used to assess the performance of textile and garment enterprises by assessing the change in total factor productivity of DMUs related to technical efficiency (CA) and technological change (FR). The results of changes in total factor productivity are shown in Table 9 and Figure 7. This index was utilized to assess the performance of 10 textile and garment enterprises. This forecasted overview can help investors make informed decisions.

Table 9 shows that the average MPI of DMUs was higher than 1 (MPI = 1.0306), meaning that total factor productivity growth in the 2017–2020 period was higher.

In the 2017–2018 period, eight out of 10 DMUs, namely, DMU1 (Viet Tien), DMU2 (Phong Phu), DMU3 (Ha Noi), DMU4 (Song Hong), DMU7 (Thanh Cong), DMU8 (TNG), DMU9 (Sai Gon), and DMU10 (TDT), achieved progress in total factor productivity. DMU4 (Song Hong) exhibited the best efficiency performance with MPI = 1.3289. Meanwhile, DMU5 (Garment 10) exhibited the worst performance with MPI = 0.8710. During this time period, the average MPI of all DMUs was 1.0843.

**Table 9.** The Malmquist productivity index of DMUs (2017–2020).

| Malmquist | 2017 ≥ 2018 | 2018 ≥ 2019 | 2019 ≥ 2020 | Average |
|---|---|---|---|---|
| DMU1 | 1.0659 | 0.9177 | 0.8540 | 0.9459 |
| DMU2 | 1.0431 | 1.0719 | 0.9321 | 1.0157 |
| DMU3 | 1.0148 | 1.0441 | 0.8397 | 0.9662 |
| DMU4 | 1.3289 | 1.2794 | 0.8800 | 1.1628 |
| DMU5 | 0.8710 | 1.1363 | 1.0376 | 1.0149 |
| DMU6 | 0.9164 | 1.0739 | 1.0239 | 1.0047 |
| DMU7 | 1.1848 | 1.0710 | 1.0183 | 1.0914 |
| DMU8 | 1.1513 | 1.0732 | 0.8629 | 1.0291 |
| DMU9 | 1.2482 | 0.8720 | 0.8180 | 0.9794 |
| DMU10 | 1.0182 | 1.1008 | 1.1697 | 1.0962 |
| Average | 1.0843 | 1.0640 | 0.9436 | 1.0306 |
| Max | 1.3289 | 1.2794 | 1.1697 | 1.1628 |
| Min | 0.8710 | 0.8720 | 0.8180 | 0.9459 |
| SD | 0.1438 | 0.1115 | 0.1141 | 0.0671 |

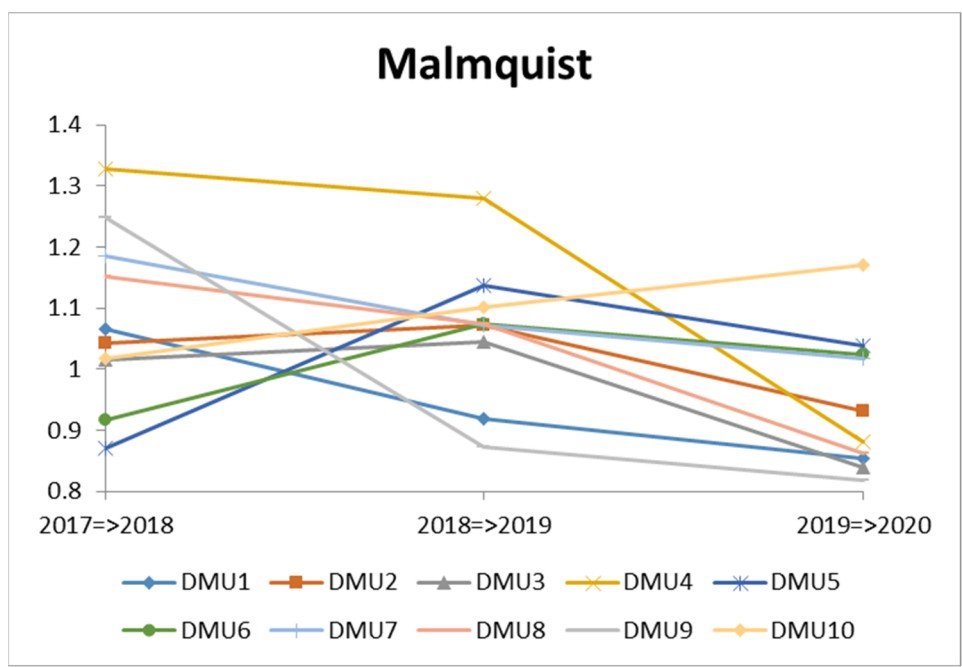

**Figure 7.** The Malmquist productivity index of DMUs (2017–2020).

In the 2018–2019 period, most DMUs achieved progress in total factor productivity. During this time period, the average MPI of all DMUs was 1.0640. The most popular DMU, DMU4 (Song Hong), exhibited the best efficiency performance with MPI = 1.2794. Meanwhile, DMU9 (Sai Gon) exhibited the worst performance with MPI = 0.8720. The

productivity of DMU5 (Garment 10) and DMU6 (Binh Thanh) improved significantly, while the productivity of the others began to decline. In particular, DMU9 (Sai Gon) had a very high MPI in the previous phase (MPI = 1.2482) but decreased to 0.8720 in this period, making it the least effective company, whereas the other enterprises were all above 0.90.

Only four DMUs (DMU5—Garment 10, DMU6—Binh Thach, DMU7—Thanh Cong, and DMU10—TDT) achieved success in total factor productivity during the 2019–2020 period, with MPI less than 1. The MPI of all DMUs dropped to 0.9436. DMU10 (TDT) exhibited the best efficiency performance with MPI = 1.1697. Meanwhile, DMU9 (Sai Gon) exhibited the worst performance with MPI = 0.8180.

Overall, DMU7 (Success) and DMU10 (TDT) had the best and most consistent results with MPI > 1 during this time.

### 4.4. Results of Epsilon-Based Measure Efficiency

The EBM model was used to rank the efficiency and inefficiency scores of 10 DMUs for the 2017–2020 in this period. The input-oriented model with a constant return to scale (EBM-I-C) was used in this study. The diversity of production possibilities set by the affinity matrix derived from the observed input and output variables was measured according to EBM protocols. Table 10 presents the input and output variables of the 10 textile and garment companies from 2017 to 2020. The maximum, minimum, average, and standard deviations (SD) were estimated by applying the statistic method. The data results are positive and acceptable for EBM model conditions.

**Table 10.** Statistics of input/output data.

| Year | | Total Assets | Cost of Goods Sold | Liabilities | Total Revenue | Gross Profit |
|---|---|---|---|---|---|---|
| 2017 | Max | 5,311,729 | 7,464,275 | 3,661,196 | 8,458,166 | 987,616 |
| | Min | 209,183 | 170,869 | 114,868 | 217,062 | 45,713 |
| | Average | 2,347,674 | 2,571,767 | 1,618,841 | 2,985,231 | 410,221 |
| | SD | 1,457,613 | 1,797,565 | 994,759 | 2,023,530 | 240,242 |
| 2018 | Max | 5,427,848 | 8,546,828 | 3,746,469 | 9,719,646 | 1,170,171 |
| | Min | 250,179 | 224,812 | 144,850 | 286,193 | 60,726 |
| | Average | 2,567,661 | 2,944,372 | 1,718,445 | 3,458,185 | 510,192 |
| | SD | 1,500,552 | 2,055,801 | 1,012,481 | 2,325,171 | 302,521 |
| 2019 | Max | 4,982,865 | 7,906,892 | 2,994,898 | 9,037,020 | 1,128,667 |
| | Min | 340,830 | 284,522 | 185,807 | 366,130 | 80,481 |
| | Average | 2,503,546 | 3,031,879 | 1,536,361 | 3,548,774 | 512,604 |
| | SD | 1,375,254 | 1,898,330 | 872,064 | 2,185,320 | 326,690 |
| 2020 | Max | 4,736,189 | 6,450,347 | 2,823,291 | 7,123,237 | 751,044 |
| | Min | 394,735 | 195,021 | 224,775 | 272,099 | 75,808 |
| | Average | 2,539,737 | 2,650,166 | 1,457,612 | 3,100,097 | 445,649 |
| | SD | 1,240,867 | 1,635,481 | 758,772 | 1,835,806 | 247,207 |

One of most important factors to consider before evaluating the effectiveness of DMUs in EBM is deciding if the data value should be positive. Furthermore, the input and output data have an isotonic relationship. The correlation coefficient, which ranges from $-1$ to 1, was used to define the relationship between two variables. If the index is close to one, the two variables have a strong relationship. If the correlation coefficient approaches 0, it indicates that the input and output are not well aligned.

The Pearson's correlation coefficients of DMUs are shown in Table 11 for each year. The correlation coefficients were higher than 0, as can be seen from the data. This suggests that all of the data variables were connected, and that EBM could be applied.

In the EBM model, the affinity index determines two factors that combine radial and non-radial models. Pearson's correlation coefficient was replaced with the affinity index between two vectors. Their appropriate value had to meet the requirement of $0 \leq P(a,b) \leq \pm 1$.

**Table 11.** Pearson's correlation coefficients.

| Year | Input/Output | Total Assets | Cost of Goods Sold | Liabilities | Total Revenue | Gross Profit |
|------|--------------|--------------|--------------------|--------------|----------------|--------------|
| | Total assets | 1 | 0.6682 | 0.9929 | 0.6555 | 0.5182 |
| | Cost of goods sold | 0.6682 | 1 | 0.6374 | 0.9991 | 0.9289 |
| 2017 | Liabilities | 0.9929 | 0.6374 | 1 | 0.6227 | 0.4706 |
| | Total revenue | 0.6555 | 0.9991 | 0.6227 | 1 | 0.9439 |
| | Gross profit | 0.5182 | 0.9289 | 0.4706 | 0.9439 | 1 |
| | Total assets | 1 | 0.7207 | 0.9910 | 0.7045 | 0.5115 |
| | Cost of goods sold | 0.7207 | 1 | 0.6836 | 0.9981 | 0.8761 |
| 2018 | Liabilities | 0.9910 | 0.6836 | 1 | 0.6629 | 0.4430 |
| | Total revenue | 0.7045 | 0.9981 | 0.6629 | 1 | 0.9040 |
| | Gross profit | 0.5115 | 0.8761 | 0.4430 | 0.9040 | 1 |
| | Total assets | 1 | 0.8359 | 0.9762 | 0.8183 | 0.6106 |
| | Cost of goods sold | 0.8359 | 1 | 0.7873 | 0.9972 | 0.8611 |
| 2019 | Liabilities | 0.9762 | 0.7873 | 1 | 0.7606 | 0.5045 |
| | Total revenue | 0.8183 | 0.9972 | 0.7606 | 1 | 0.8965 |
| | Gross profit | 0.6106 | 0.8611 | 0.5045 | 0.8965 | 1 |
| | Total assets | 1 | 0.8137 | 0.9503 | 0.8098 | 0.6328 |
| | Cost of goods sold | 0.8137 | 1 | 0.8086 | 0.9967 | 0.7921 |
| 2020 | Liabilities | 0.9503 | 0.8086 | 1 | 0.7922 | 0.5333 |
| | Total revenue | 0.8098 | 0.9967 | 0.7922 | 1 | 0.8389 |
| | Gross profit | 0.6328 | 0.7921 | 0.5333 | 0.8389 | 1 |

The deviation of variables was used to calculate the diversity index of vectors and $0 \leq D(a,b) = D(b,a) \leq \frac{1}{2}$. When two vectors are proportional, it is only equal to 0. To ensure that the correspondence of input and output variables was acceptable for measuring the effectiveness of DMUs in EBM, both affinity and diversity indicators were used.

Tables 12 and 13 show the EBM model's diversity index and affinity index matrices for the years 2017–2020, respectively.

The diversity and affinity matrices had values ranging from 0 to 0.2807 and from 0.4387 to 1, respectively, according to the results. These values satisfy the model's requirements. As a result, EBM could be used to rank the efficiency and inefficiency of DMUs.

To eliminate the EBM score with each DMU, the input/output weighting and epsilon quotient are essential. The ratio of input to output was determined using a weighted index. The complete weighted indices were positive, according to Table 14. Thus, in this case, changes in the input would have an effect on the output, whereby, if the input value increased, the output value would also increase accordingly.

**Table 12.** Diversity matrix in EBM model (2017–2020).

| Period | Input | Total Assets | Cost of Goods Sold | Liabilities |
|---|---|---|---|---|
| **2017** | Total assets | 0 | 0.2769 | 0.1711 |
| | Cost of goods sold | 0.2769 | 0 | 0.2807 |
| | Liabilities | 0.1711 | 0.2807 | 0 |
| **2018** | Total assets | 0 | 0.2230 | 0.1448 |
| | Cost of goods sold | 0.2230 | 0 | 0.2480 |
| | Liabilities | 0.1448 | 0.2480 | 0 |
| **2019** | Total assets | 0 | 0.1382 | 0.2118 |
| | Cost of goods sold | 0.1382 | 0 | 0.1790 |
| | Liabilities | 0.2118 | 0.1790 | 0 |
| **2020** | Total assets | 0 | 0.1467 | 0.2520 |
| | Cost of goods sold | 0.1467 | 0 | 0.1722 |
| | Liabilities | 0.2520 | 0.1722 | 0 |

**Table 13.** Affinity matrix in EBM model (2017–2020).

| Period | Input | Total Assets | Cost of Goods Sold | Liabilities |
|---|---|---|---|---|
| **2017** | Total assets | 1 | 0.4461 | 0.6579 |
| | Cost of goods sold | 0.4461 | 1 | 0.4387 |
| | Liabilities | 0.6579 | 0.4387 | 1 |
| **2018** | Total assets | 1 | 0.5541 | 0.7104 |
| | Cost of goods sold | 0.5541 | 1 | 0.5041 |
| | Liabilities | 0.7104 | 0.5041 | 1 |
| **2019** | Total assets | 1 | 0.7237 | 0.5765 |
| | Cost of goods sold | 0.7237 | 1 | 0.6421 |
| | Liabilities | 0.5765 | 0.6421 | 1 |
| **2020** | Total assets | 1 | 0.7065 | 0.4961 |
| | Cost of goods sold | 0.7065 | 1 | 0.6556 |
| | Liabilities | 0.4961 | 0.6556 | 1 |

**Table 14.** Epsilon and weight for input/output of the EBM model (2017–2020).

| Period | Total Assets | Cost of Goods Sold | Liabilities |
|---|---|---|---|
| **2017** | 0.3510 | 0.2993 | 0.3497 |
| **2018** | 0.3496 | 0.3090 | 0.3414 |
| **2019** | 0.3349 | 0.3453 | 0.3198 |
| **2020** | 0.3281 | 0.3541 | 0.3178 |

The results of Epsilon for EBM over the years in Table 15 satisfied the condition: $0 \leq$ Epsilon index $\leq 1$.

**Table 15.** Epsilon for EBM every year from 2017 to 2020.

| Year | Epsilon Indicator |
|------|-------------------|
| 2017 | 0.4821 |
| 2018 | 0.4082 |
| 2019 | 0.3517 |
| 2020 | 0.3785 |

For EBM, the efficiency of 10 textile and garment enterprises was calculated using the factor weight and epsilon. From 2017 to 2020, the efficiency scores for DMUs were as shown in Table 16. Overall, all companies had high productivity, with no company having an efficiency score below 0.7141.

**Table 16.** Efficiency scores of EBM model from 2017 to 2020.

| Symbol | DMUs | 2017 | 2018 | 2019 | 2020 |
|--------|------|------|------|------|------|
| Viet Tien | DMU1 | 0.9733 | 1 | 0.9722 | 0.8987 |
| Phong Phu | DMU2 | 0.7141 | 0.7299 | 0.7571 | 0.7709 |
| Ha Noi | DMU3 | 0.7742 | 0.7860 | 0.7806 | 0.7769 |
| Song Hong | DMU4 | 0.9500 | 1 | 1 | 1 |
| Garment 10 | DMU5 | 1 | 0.9382 | 1 | 1 |
| Binh Thanh | DMU6 | 0.9517 | 0.8773 | 0.8871 | 0.9427 |
| Thanh Cong | DMU7 | 0.8544 | 0.8914 | 0.8905 | 0.9346 |
| TNG | DMU8 | 0.8992 | 0.8992 | 0.9179 | 0.8909 |
| Sai Gon | DMU9 | 1 | 1 | 0.9748 | 0.9034 |
| TDT | DMU10 | 1 | 0.9986 | 0.9635 | 1 |

As reported in Table 16, there were three DMUs with efficiency scores that increased over time: DMU2 (Phong Phu), DMU4 (Song Hong), and DMU7 (Thanh Cong). In contrast, the other DMUs showed a downward trend.

In particular, DMU4 (Song Hong) and DMU5 (Garment 10) obtained a strong efficiency. DMU4 (Song Hong) was the company having the best performance over the years despite starting with a score of 0.9500 in 2017; however, in the next 3 years, it jumped suddenly and retained its top ranking with theta always equal to 1 and slacks of 0. DMU5 (Garment 10) started well with a score of 1 (in 2017) but suddenly dropped to 0.9382 in 2018; from 2019 to 2020 it achieved the top rank and continuously maintained the same level with a score of 1.

DMU10 (TDT) presented a high performance but was not stable. In 2017, it had a pretty good rank at 1 with a score of 1. However, 2 years later (2018–2019) it was reduced to 0.9985 and 0.9634, ranking third. In 2020, it returned to the rank of first.

DMU2 (Phong Phu), DMU3 (Ha Noi), DMU6 (Binh Thanh), DMU7 (Thanh Cong), and DMU8 (TNG) showed a weak performance score. In particular, DMU2 (Phong Phu) and DMU3 (Hanoi) had the lowest efficiency score across the whole period considered from 2017 to 2020. Their scores declined constantly, and their ranks dropped, remaining at the bottom of the ranking.

Figure 8 illustrates the comparison of DMU efficiency scores. As can be seen from the chart, the performance score among companies differed in each period of time.

DMU4 (Song Hong), DMU5 (May 10), and DMU10 (TDT) were rated as the three most effective units and ideal suppliers, achieving a rank of first and maintaining it over time. In contrast, DMU2 (Phong Phu) and DMU3 (Hanoi) tended occupy the worst positions.

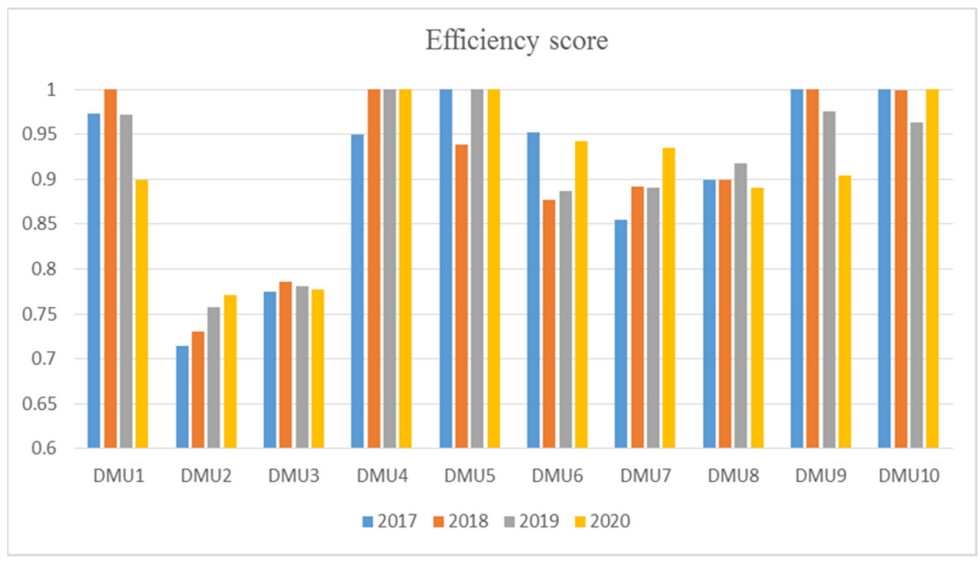

**Figure 8.** Ranking of 10 textile and garment companies.

## 5. Discussion and Conclusions

### 5.1. Discussion

The results presented above show a picture of Vietnam's textile and garment industry in recent years. Figure 9 shows the MIP of the average ranking of DMUs from 2017 to 2020. Overall, most DMUs performed well (MPI > 1). DMU4 (Song Hong) exhibited the best performance. On the other hand, only three DMUs (DMU9 (Sai Gon), DMU3 (Ha Noi), and DMU1 (Viet Tien)) did not exhibit efficient performances. DMU9 (Sai Gon) and DMU3 (Ha Noi) had slight fluctuations in the two periods 2018–2019 and 2019–2020, leading to ineffective results. Although DMU3 was effective in the two periods 2017–2018 and 2018–2019, a sharp decline in the last stage resulted in an inefficient average total. This means that these three enterprises need to focus on improving their technical and technological efficiency every year.

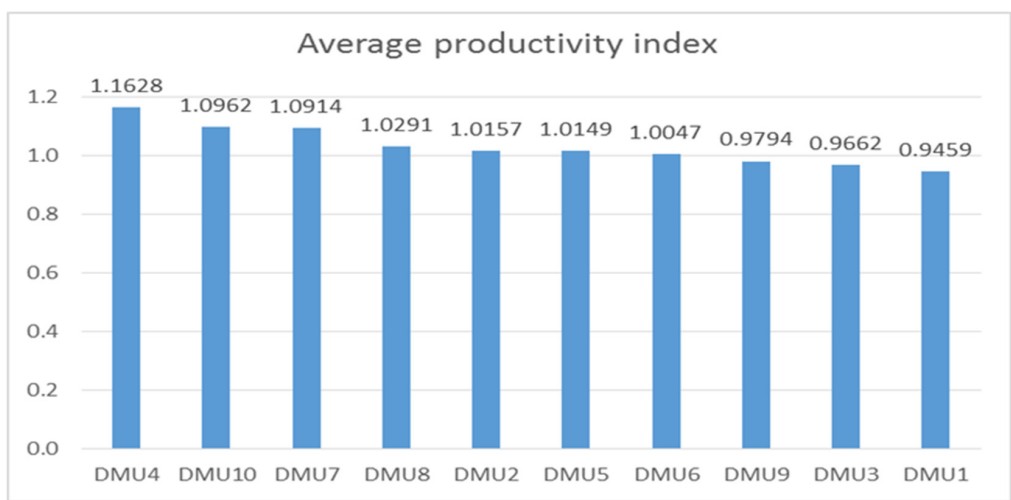

**Figure 9.** The average Malmquist indices of decision-making units (DMUs) for the period 2017–2020.

In particular, 2020 was a very difficult year for Vietnam's textile and garment industry, affected by the COVID-19 epidemic. According to the financial report in Vietnam, in 2020, Song Hong Garment Joint Stock Company (DMU4) revenues decreased by 20%, and profit decreased by 60% compared to 2019 [51]. With a similar situation, Garment Corporation 10 Joint Stock Company (DMU5) revenues were only 81% of those the previous year. In the

same context of the COVID-19 epidemic, growth increased rapidly due to the promotion of masks and protective clothing. As a result, the company's net profit increased by 28%.

Overall, textile and garment enterprises should have policies to improve competitiveness in terms of quality, as well as actively diversify the supply of raw materials, auxiliary materials, etc. They should also be flexible in their production and business plans. Many businesses have found some opportunities in niche markets such as masks and specialized protective clothing, not only serving the domestic market but also exporting to the international market.

*5.2. Conclusions*

This study evaluated the performance of 10 Vietnamese textile and garment enterprises from 2017 to 2020 using the DEA–Malmquist index and the EBM model to provide an overview of the Vietnamese textile and garment industry. Total assets, cost of goods sold, and liabilities were the inputs in the study, while total sales and gross profit were the output variables. The purpose of the study was to give an overview of the Vietnamese textile and garment industry through focusing on technical efficiency change, technological change, and overall factor productivity, as well as the efficiency and inefficiency of businesses. The Malmquist–DEA model was used in the study to create a picture of the productivity growth of companies from 2017 to 2020. On the other hand, the EBM model utilized ratings to calculate performance and underperformance scores to give recommendations for incompetent firms to improve their efficiency. As a result, the combination of the two methodologies created a more efficient and equitable framework for evaluating the performance and growth of companies from all perspectives.

The primary contributions of this study are as follows: (1) this study presented a method to assess the textile and garment industry that combines the DEA–Malmquist and EBM models. This hybrid technique was first introduced in the study to assess the performance of the textile and garment industry, following a literature review. This combination could measure the total relative productivity of DMUs in multiple stages, with multiple input and output variables through engineering and technology evaluation. On the other hand, it was possible to evaluate the efficiency/inefficiency conditions of the DMUs, while also considering the proportional changes of the input/output. (2) The results of this study provide thorough and factual information on Vietnam's top 10 textile and garment enterprises in recent years. Managers and policymakers can find ways to assure a stable supply of resources and quality for enterprises by adapting production and company growth plans, as well as the possibility of changing the business situation. (3) The authors expect that the results of this study can reflect the present state of the textile and garment industry through the activities of some of the leading textile and garment companies in Vietnam. Therefore, this study can be a useful guide for decision makers, investors, and consumers who are looking to improve their performance toward sustainable development, not just in Vietnam but worldwide.

Although the research was successful, there were undeniable limitations. Firstly, this study only evaluated at the performance of 10 Vietnamese textile and garment companies. More companies should be included for a more detailed overview. This was due to a lack of annual reports. Future studies can further analyze textile companies to get a more realistic picture of Vietnam's textile and garment activities using different methods, in order to improve competitiveness and maintain an important position in the textile industry. Secondly, future studies also need to choose more input and output variables for further comparison in order to increase the objectivity of the research. Thirdly, this study only focused on a particular industry in particular country; hence, there may have been some bias related to the supplier selection criteria in the textile and garment industry.

**Author Contributions:** Conceptualization, P.-T.T.N.; data curation, P.-T.T.N.; formal analysis, Y.-H.W.; funding acquisition, C.-N.W.; investigation, Y.-H.W.; methodology, P.-T.T.N.; project administration, C.-N.W.; software, P.-T.T.N.; validation, T.-T.D.; writing—original draft, P.-T.T.N.; writing—review and editing, T.-T.D. All authors read and agreed to the published version of the manuscript.

**Funding:** This research received no external funding.

**Institutional Review Board Statement:** Not applicable.

**Informed Consent Statement:** Not applicable.

**Data Availability Statement:** Not applicable.

**Acknowledgments:** The authors appreciate the support from the National Kaohsiung University of Science and Technology, Taiwan and Hong Bang International University, Vietnam.

**Conflicts of Interest:** The authors declare no conflict of interest.

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
