# Peer review of "A Study of Performance Evaluation for Textile and Garment Enterprises"

_processes, doi:10.3390/pr10112381_

Round 1
Reviewer 1 Report
Dear Authors,
The study is quite interesting and has a scientific soundness.
However, there are a few parts that need extra attentions and actions from all of you, especially,
1) In term of citations for the methods part (e.g. Pearson Correlation, DEA Malmquist model, proper sources (citations) are not included to indicate your references. Hence, it made me wonder, are claiming all the methods as yours? If those methods are not yours, then it will raise an ethical concern, at least to me, and should be to the journal as well.
2) Other comments of mine, please refer to the attachment (Comments for the Reviewer) given. Please take actions accordingly.
Thank you.

Author Response
Thanks for your valuable comments on the previous manuscript. According to these comments, we have revised and improved the manuscript carefully. Our reply to the comments is listed as follows:
Best regards,
Phuong-Thuy Thi Nguyen et al.

Reviewer 2 Report
In this work, the authors have evaluated the performance of 10 textile and garment companies in Vietnam from 2017 to 2020 by combining Data Envelopment Analysis (DEA) Malmquist and an Epsilon-Based Measure (EBM) model. The Malmquist model has been applied to assess the total productivity growth rates of the companies based on technical efficiency change (catch-up index) and technological change (frontier-shift index). And the EBM model has been used to calculate the performance and inefficiency scores, and the ratings of each company.
Overall, the paper is somewhat interesting, and there are some new results in the paper. The paper needs a minor revision before acceptance for publication.
1. The abstract should be improved carefully by mentioning the resulting advantages of the presented approach.
2. The introduction section should be updated with the motivation behind this study, research gap, novelty, and contributions of the work. What are technical contributions?
3. Add some graphs to represent the data related to Vietnam's textile and garment industry for more understanding to the readers.
4. Why have you selected the Pearson Correlation coefficient over others?
5. In subsection 3.4.2, what is representing d? In some expressions, 'd 'is italic, and some have a simple ‘d’?
6. Improve the results discussion part of the paper.
7. What characteristics of the adopted method make it a superior fit for the problem considered in this manuscript?
Author Response

(The authors gave the same response as above.)

Round 2
Reviewer 1 Report
Dear authors,
Thank you for your actions in responding to my first round of comments.
Overall, I quite satisfied with the responses given.
However, I am still found that the numbering label of Figures and Tables were still NOT corrected accordingly as in my previous comments. There are still several mistakes found on the arrangements of the Figures and Tables.
Examples: (Please check others too)
Table 1 (Page 6), Table 1 (Page 9)
Table 2 (Page 7), Table 2 (Page 14)
Please pay a careful attentions as it will affect the quality of the paper that will be published by the journal.
Please double check the numbering labels of ALL FIGURES and TABLES and repair accordingly.
Please make sure that all the Figure/Table numbers are NOT repeated.
Thank you.
Author Response
Thank you for the very helpful review. We have checked and revised all the order of Tables and Figures for the whole text.
Thank you.
